# Enhancing laser beam performance by interfering intense laser beamlets

A. Morace[1], N. Iwata[1], Y. Sentoku[1], K. Mima[1], Y. Arikawa[1], A. Yogo[1], A. Andreev[2,3], S. Tosaki[1], X. Vaisseau[1], Y. Abe [1], S. Kojima [1], S. Sakata [1], M. Hata[1], S. Lee [1], K. Matsuo [1], N. Kamitsukasa[1], T. Norimatsu[1], J. Kawanaka [1], S. Tokita[1], N. Miyanaga[1], H. Shiraga[1], Y. Sakawa[1], M. Nakai[1], H. Nishimura[1], H. Azechi[1], S. Fujioka [1] & R. Kodama[1]

Increasing the laser energy absorption into energetic particle beams represents a long-standing quest in intense laser-plasma physics. During the interaction with matter, part of the laser energy is converted into relativistic electron beams, which are the origin of secondary sources of energetic ions, γ-rays and neutrons. Here we experimentally demonstrate that using multiple coherent laser beamlets spatially and temporally overlapped, thus producing an interference pattern in the laser focus, significantly improves the laser energy conversion efficiency into hot electrons, compared to one beam with the same energy and nominal intensity as the four beamlets combined. Two-dimensional particle-in-cell simulations support the experimental results, suggesting that beamlet interference pattern induces a periodical shaping of the critical density, ultimately playing a key-role in enhancing the laser-to-electron energy conversion efficiency. This method is rather insensitive to laser pulse contrast and duration, making this approach robust and suitable to many existing facilities.

[1] Institute of Laser Engineering, Osaka University, Suita 565-0871, Japan. [2] Max Born Institute for non-linear optics and short pulse spectroscopy, Berlin 12489, Germany. [3] St. Petersburg State University, Sankt-Petersburg 199034, Russia. Correspondence and requests for materials should be addressed to A. M. (email: morace@ile.osaka-u.ac.jp)

The recent development of multi-kJ and multi-PW laser systems typically coupled to large laser facilities represented a major step for high energy density (HED) physics research. These laser systems are devoted to the generation of bright particle and X-ray sources that find a myriad of applications in HED science, from isochoric heating of materials or dense plasmas to determine equations of state[1–3] to X-ray or proton radiography of HED plasmas[4,5] as well as generation of collisionless shocks for experimental astrophysics[6].

Within the last decade three major multi-kJ PW-class laser systems have been developed and commissioned: LFEX laser at the Institute of Laser Engineering, Osaka University, Japan[7], the ARC laser at the National Ignition Facility, Lawrence Livermore National Laboratory, U. S.[8] and Petal laser system on the Laser MagaJoule at the Commissariat a' l'Energie Atomique in France[9]. With pulse energies exceeding the kJ level, a single beamlet laser with picosecond pulse duration requires the design of very large and expensive optics in particular regarding diffraction gratings for pulse compression. These lasers are therefore all composed by multiple beamlets generated from a single oscillator and subsequently divided, undergoing amplification and compression separately and then focused back together on target by a single (LFEX and Petal lasers) or multiple off-axis parabolas (ARC laser). Initially considered as a necessary laser system design to have a multi-kJ, picosecond laser sources, the multi-beamlet irradiation on LFEX laser revealed to be an important tool to study advanced laser-plasma interaction (LPI) regimes either by arranging the beamlets in temporal sequence[10,11] or temporally and spatially overlapped.

One of the most reliable diagnostics for laser energy absorption in LPI is represented by target normal sheath accelerated (TNSA) proton beams[12–19]. As TNSA mechanism relies on the energy transfer from fast electron to ions through large charge separation electric fields at the plasma-vacuum interface, the ion beam performances are directly dependent on the fast electron generation via LPI.

A widely adopted approach to optimize laser energy absorption through LPI is target structuring by deposition of foams[20], micro or nanoparticles[21–23], microwires[24], by engraving gratings mechanically or by pre-imprinting laser interference patterns on the target surface[25–27]. Although presenting clear advantages, these techniques generally require very high laser contrast in order to prevent the destruction of the nanostructures by the laser pedestal and have been demonstrated mainly using femtosecond pulses from titanium-sapphire (Ti:Sa) type lasers. Recently the generation of transient plasma gratings using two interfering pre-pulses was demonstrated[28], with several potential applications to LPI especially with ultra-short Ti:Sa laser systems.

In this work we introduce a completely different approach to laser-plasma interaction that allows to significantly improve the laser energy absorption into hot electrons with consequent TNSA enhancement, consisting in focusing multiple coherent beamlets spatially and temporally overlapped and therefore undergoing self-induced interference in the common focal position. The resulting interference pattern, which periodicity depends on the incidence angle between the beamlets, will induce a critical surface modulation via radiation pressure, strongly improving the laser energy absorption into hot electrons during the intense laser irradiation itself, in a fashion similar to micro/nanostructured targets but in a totally self-consistent manner, without need for special target preparation or high-contrast operation and suitable for picosecond pulse duration.

## Results

**Experimental results**. The proof of principle was demonstrated on LFEX laser, composed by four coherent beamlets focused by a single off-axis parabola and capable of delivering up to 500 J (~350 J after compression) per beamlet at peak compression (~1.5 ps) onto an ~60 μm focal spot area, resulting in a nominal intensity of ~$1 \times 10^{19}$ W/cm$^2$ ($a_0$ ~3) as shown in Fig. 1a. Each beamlet incidence angle with respect to the parabola's optical axis $\vartheta_i$ is 2.6 degrees, resulting in a mosaic interference pattern of wavelets disposed with periodicity $P$ of 11 μm according to the following relation $P = \lambda / 2 \sin \vartheta_i$ (an acquisition of LFEX focal spot is shown in Fig. 1b).

Thin 5 μm Aluminum (Al) foils were irradiated with 1 and 4 beamlets keeping a constant total laser energy of 270 J on target and therefore constant nominal intensity of ~$2.5 \times 10^{18}$ W/cm$^2$. Shot-to-shot energy fluctuations amounted for only 7% of the total laser energy. LPI-generated fast electrons and TNSA proton beams were recorded, respectively, by magnetic electron spectrometer with minimum resolved energy of 1 MeV and Thomson parabola with minimum detected proton energy of 6 MeV, as shown in the schematic in Fig. 1a.

The Al foils were positioned normally to the LFEX parabola's optical axis, Thomson parabola and electron spectrometer, both looking at the target rear side, were mounted, respectively, collinearly and 20.9 degrees from the parabolas optical axis.

Experimental results (see Fig. 1c, d) show clear influence of beamlet interference on both fast electron and ion generation. Electron spectrometer data for 4 interfering beamlets irradiation show hotter fast electron slope temperature and significantly higher laser-to-electron energy conversion efficiency compared to the single beamlet case, with electron temperatures respectively of 0.6 MeV and 0.36 MeV and a 2.9 fold increment in conversion efficiency for electron energy exceeding 1 MeV.

Thomson parabola data show higher peak proton energy and about 3.1 fold increment in laser-to-proton energy conversion efficiency for interfering beamlets case compared to single beamlet irradiation.

**Numerical modeling**. To explain the experimental data and understand the LPI for interfering beamlets we carried out two-dimensional (2D) collisionless particle-in-cell (PIC) simulations using the Epoch code[29]. The first set of simulations is focused on modeling the interaction for conditions close to the experimental ones. The target was modeled as a 5 μm hydrogen layer with a pre-formed plasma extending from the front surface for 5 μm with scale-length of 1 μm. The problem was reduced to 2D geometry due to computational limitation using 2-beamlets interference, each with an intensity of $1.25 \times 10^{18}$ W/cm$^2$ and comparing the results with single beamlet interaction with an intensity of $2.5 \times 10^{18}$ W/cm$^2$. The beamlets incidence angle, spot size and total nominal intensity correspond to those of LFEX laser in the experiment. The simulations closely reproduce the experimental results with larger laser-to-electron and laser-to-ion energy conversion efficiency as well as higher hot electron slope temperature and peak proton energy for interfering beamlets compared to single beamlet irradiation as represented in Fig. 1e, f. In particular the hot electron slope temperature increases by a factor 1.5 while the laser-to-electron energy conversion efficiency by 2.7 for electrons exceeding 1 MeV (while an overall 2-fold increment is recorded for electrons with energy ≥100 keV), very close to the experimental values showing a temperature increment of 1.6 and overall conversion efficiency gain of 2.9 for energies exceeding 1 MeV. Correspondingly TNSA proton data show higher peak proton energy with an overall laser-to-proton conversion efficiency increment of 2.3 fold for interfering beamlets, while considering only the protons with energies exceeding 6 MeV the conversion efficiency increases by 3.1 times, very close to the experimental value of 3.3.

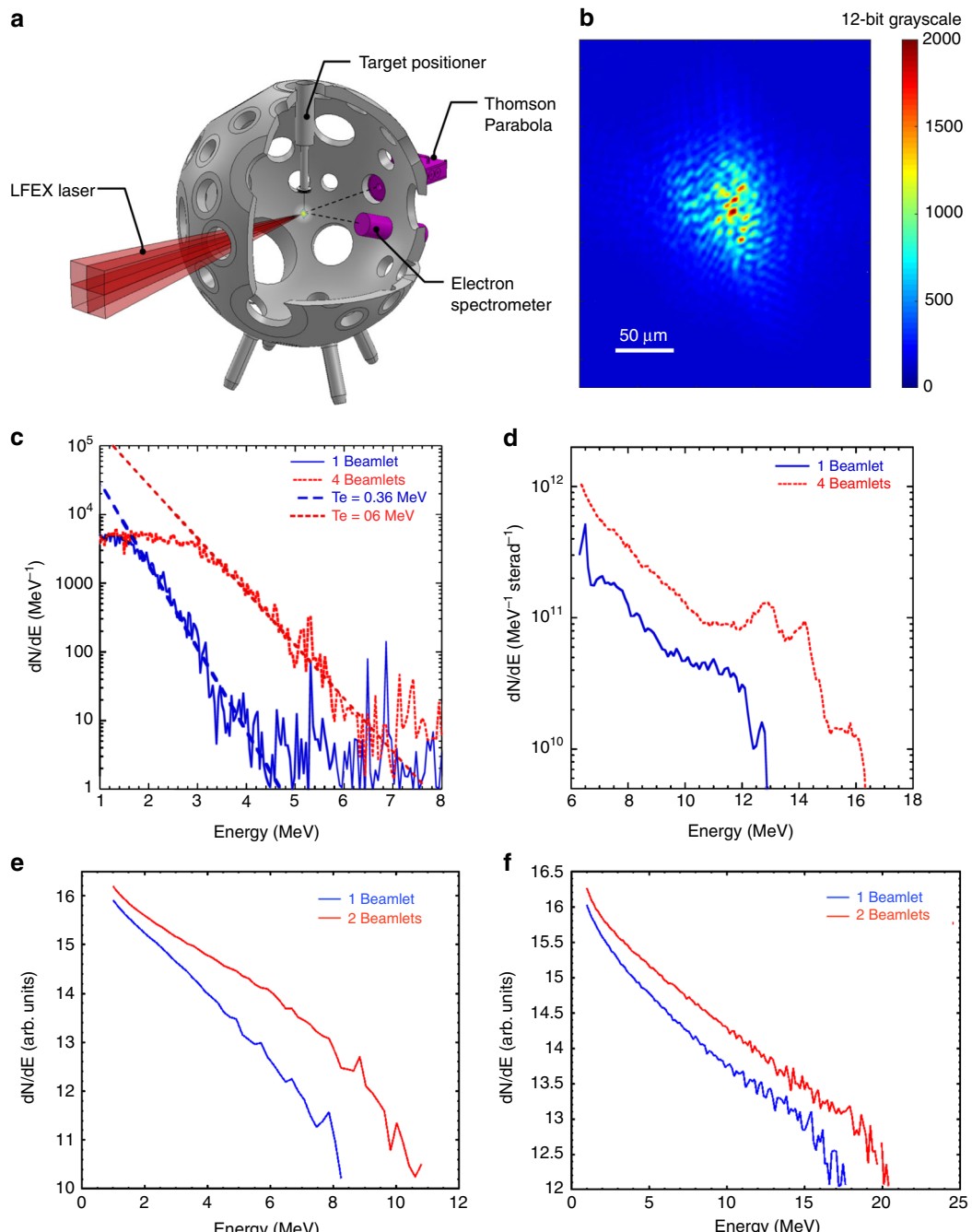

**Fig. 1** Experimental setup and results. **a** Schematic of the experimental setup. The energy as well as the nominal intensity on target are kept constant and correspond respectively to 270 J and $2.5 \times 10^{18}$ W/cm². Fast electron and proton beam data are collected by electron spectrometer and Thomson parabola. **b** Far field image of the LFEX focal spot using the ~ 1mJ OPCPA pulse showing the 4-beamlet interference pattern. **c**, **d** Experimental results for fast electron spectra and proton spectra, respectively. The red dashed line represents the four-beamlet interaction data and the blue solid line the single beamlet interaction data. It appears clear the enhancement of fast electron and proton beam generation for interfering beamlets. 2D PIC simulation results for **e** fast electron and **f** proton beam generation for laser configuration close to the experimental one. The red line represents the 2-beamlet irradiation result and the blue line the single beamlet one

We can identify three fundamental effects that significantly differ from the single beamlet interaction: wavelet self-focusing in the underdense plasma, critical surface modulation induced by the interference pattern and very large surface magnetic field generation in the LPI region (See Fig. 2a–c).

In order to understand how these effects play a role in the hot electron generation we performed a set of simulations using laser intensity of $3 \times 10^{18}$ W/cm² ($a_0 = 1.5$) for different beamlets

incidence angles $\vartheta_i$ ranging from 2.6 degrees to 25 degrees corresponding to interference period $P$ from 11 μm to 1.18 μm as summarized in Table 1.

We focus our attention here on the fast electron beam generation, ultimately responsible for the TNSA proton acceleration and we compare the results with single beamlet irradiation at normal incidence with the same total nominal intensity as the two beamlets combined. The adoption of a normally incident single

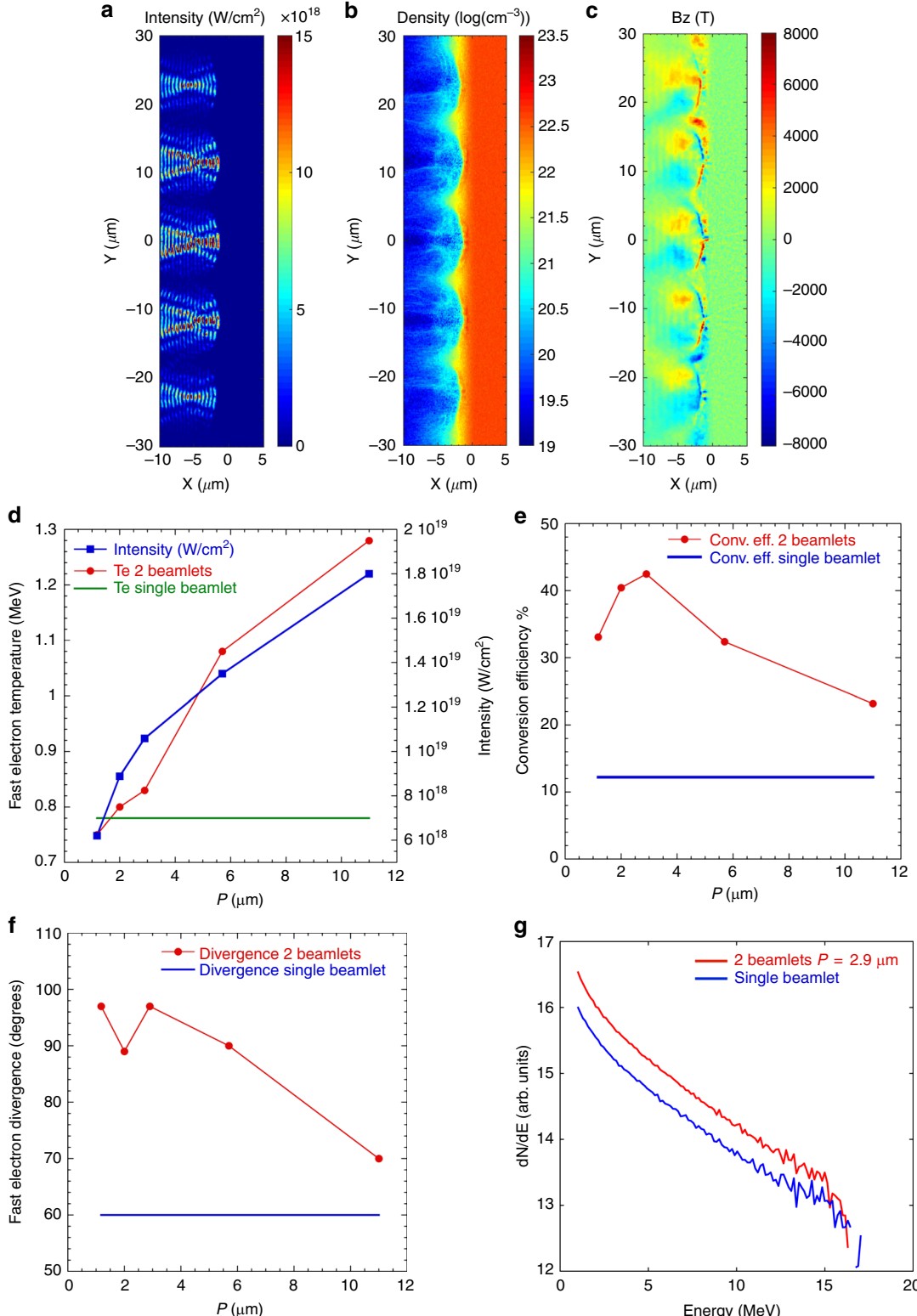

beam is based on the fact that during interference the initial incidence angle only determines the interference period and each single wavelet interacts with the target as normally incident.

**Wavelets self-focusing.** The hot electron energy spectra show a clear dependence on the interference. For large periods the hot electron temperature is substantially higher compared to the

single beam case and steadily decreases for smaller periods reaching approximately the same temperature as for the single beam as shown in Fig. 2d. Undergoing interference the beamlets are decomposed into wavelets according to the interference pattern, facilitating the self-focusing for each wavelet in the expanding underdense plasma thus increasing the laser intensity on target. As the wavelet size decreases with the period,

**Fig. 2** Simulation results for different values of the interference period P. Simulation snapshots at peak laser intensity for $P = 11\,\mu m$ displaying **a** the laser intensity distribution in the near-critical density plasma clearly showing wavelet self-focusing, **b** plasma electron density showing significant surface modulation induced by wavelets hole-boring, and **c** the formation of large surface magnetic fields in the LPI region. **d** Simulation results for fast electron temperature (red dotted line) and laser intensity measured at the critical surface (blue dotted line) as function of interference period for 2-beamlets irradiation and fast electron temperature for single beamlet irradiation (green straight line). The single beamlet data covers all periods to visually facilitate the comparison with the 2-beamlets case. As the period reduces so does the fast electron temperature approaching the value for single beamlet irradiation. **e** Laser-to-electron energy conversion efficiency as function of interference period for 2-beamlet (red dotted line) and single beamlet (blue straight line) irradiation. The single beamlet line covers all periods to visually facilitate the comparison with the interfering beamlets case. The conversion efficiency for 2-beamlets irradiation is higher by factors compared to the one for single beamlet for all values of P. **f** Full-angle fast electron divergence as function of interference period for 2-beamlet (red dotted line) and single beamlet (blue straight line) irradiation. Multi-beamlet irradiation presents larger fast electron divergence than the single beamlet case and the divergence angle rapidly saturates to values between 90 and 100 degrees for $P \leq 5.9\,\mu m$. **g** Proton energy spectra for 2 beamlet with $P = 2.9\,\mu m$ (red line) and single beamlet interaction (blue line). Although the peak proton energy remains substantially unchanged for the two cases, the laser-to-proton energy conversion efficiency for interfering beamlets is 2.9 times higher than for single beamlet case

**Table 1 List of beamlets incidence angle $\vartheta_i$ and correspondent value of P**

| Beamlet incidence angle $\vartheta_i$ | Interference period P |
|---|---|
| 2.6 degrees | 11 μm |
| 5 degrees | 5.7 μm |
| 10 degrees | 2.9 μm |
| 14.5 degrees | 2 μm |
| 25 degrees | 1.18 μm |

For larger incidence angles the wavelet size (~60% of P at full width at half maximum) reduces below the laser wavelength for 25 degrees incidence angle case

self-focusing reduces as confirmed by the amplitude of the laser intensity at the plasma surface (Fig. 2d) with the hot electron temperature approaching the values corresponding to single beam irradiation.

In the ideal case of four interfering beamlets, the peak intensity at the maxima should increase by a factor 4 compared to the nominal or average intensity. However, this increment in peak intensity does not constitute the major factor determining the hot electron temperature given its dependence on the interference period P.

**Surface modulation**. The laser-to-electron energy conversion efficiency for interfering beamlets increases by factors compared to the single beamlet case for all interference periods and presents a maximum for $P = 2.9\,\mu m$ as shown in Fig. 2e.

To understand how the beamlet interference influences the laser-to-electron energy conversion efficiency we focus our attention on the plasma electron density profile $n_e$ between 1.5 $n_c$ and 6 $n_c$ ($n_c$ being the critical density for 1.054 nm light) taken at simulation time corresponding to peak laser intensity on target as represented in Fig. 3a for single and multi-beamlet irradiation for different values of P. For single beamlet irradiation the electron density profile is rather uniform while for the 2-beamlet case we observe a significant surface modulation with periodicity following the interference pattern generated by the wavelets hole-boring pressure. We define an average surface modulation angle $\alpha$, corresponding to the average wavelet incidence angle on target as shown in Fig. 3b. The critical surface modulation plays a key role in the enhancement of laser-to-electron conversion efficiency by dynamically increasing the average wavelet incidence angle on target and then the fraction of laser energy absorbed by vacuum heating-type mechanism. As can be observed in Fig. 3a, the modulation depth remains approximately constant for $P \geq 2.9\,\mu m$ and reduces as the period further decreases. As the period approaches the laser wavelength the wavelet size becomes comparable to the electron excursion length in the laser field, therefore preventing the wavelet from further penetrating

through hole-boring in the overdense plasma. As consequence the modulation angle $\alpha$ increases as the period decreases reaching its maximum at $P = 2.9\,\mu m$ and then reduce again for smaller periods, following the same trend as the one for conversion efficiency described before.

The energy integrated fast electron beam divergence for interfering beamlets is also higher compared to the single beamlet for all cases and from a minimum of 70 degrees rapidly rises and stabilizes to values between 90 and 100 degrees as the P decreases as represented in Fig. 2f. Figure 2g shows the proton energy spectrum obtained in the simulation for $P = 2.9\,\mu m$. Substantially no change is observed in maximum proton energy while an overall 2.8 fold increment in conversion efficiency for energy above 1 MeV is obtained for 2-beamlet irradiation compared to the single beamlet case. This result can be explained considering that despite the laser-to-electron conversion efficiency is maximized, this is mitigated by the significant increase in the hot electron divergence up to about 100 degrees full-angle while the hot electron temperature does not sensibly change from the single beamlet case. In these conditions the amplitude of the accelerating electric field remains substantially unchanged and so the peak proton energy while the larger spatial extension of the electric field guarantees the acceleration of a larger number of protons, therefore increasing the overall laser-to-protons energy conversion efficiency.

**Surface magnetic field**. By increasing the laser incidence angle for p-polarized light, the component of the fast electron current density parallel to the target surface $J_\parallel$ also increases. This together with the enhanced laser energy absorption by vacuum heating generates a large surface B-field which prevents part of the laser generated fast electrons from penetrating in the overdense plasma. The fraction of deflected electrons flows along the surface, confined by the balanced action between the charge separation electric field and the surface B-field, further enhancing $J_\parallel$ and consequently the B-field itself in a self-consistent manner.

The maximum laser energy conversion efficiency in fast electrons injected into the target material is expected to saturate for incidence angles between 30 and 40 degrees for p-polarized laser light[30]. For incidence angles on the order of 50 degrees as much as 50% of the fast electron population is prevented from penetrating in the overdense plasma at any distance from the laser focus and this fraction goes up to 100% for inclination exceeding a critical angle at about 70 degrees[31].

In our model a striking linear correlation is found between the laser to electron energy conversion efficiency and the modulation angle $\alpha$ as represented in Fig. 3c, steadily increasing even for values of $\alpha$ exceeding 40 degrees. The explanation for this behavior is found investigating the origin and role of the large surface magnetic fields generated for multi-beamlet irradiation. In

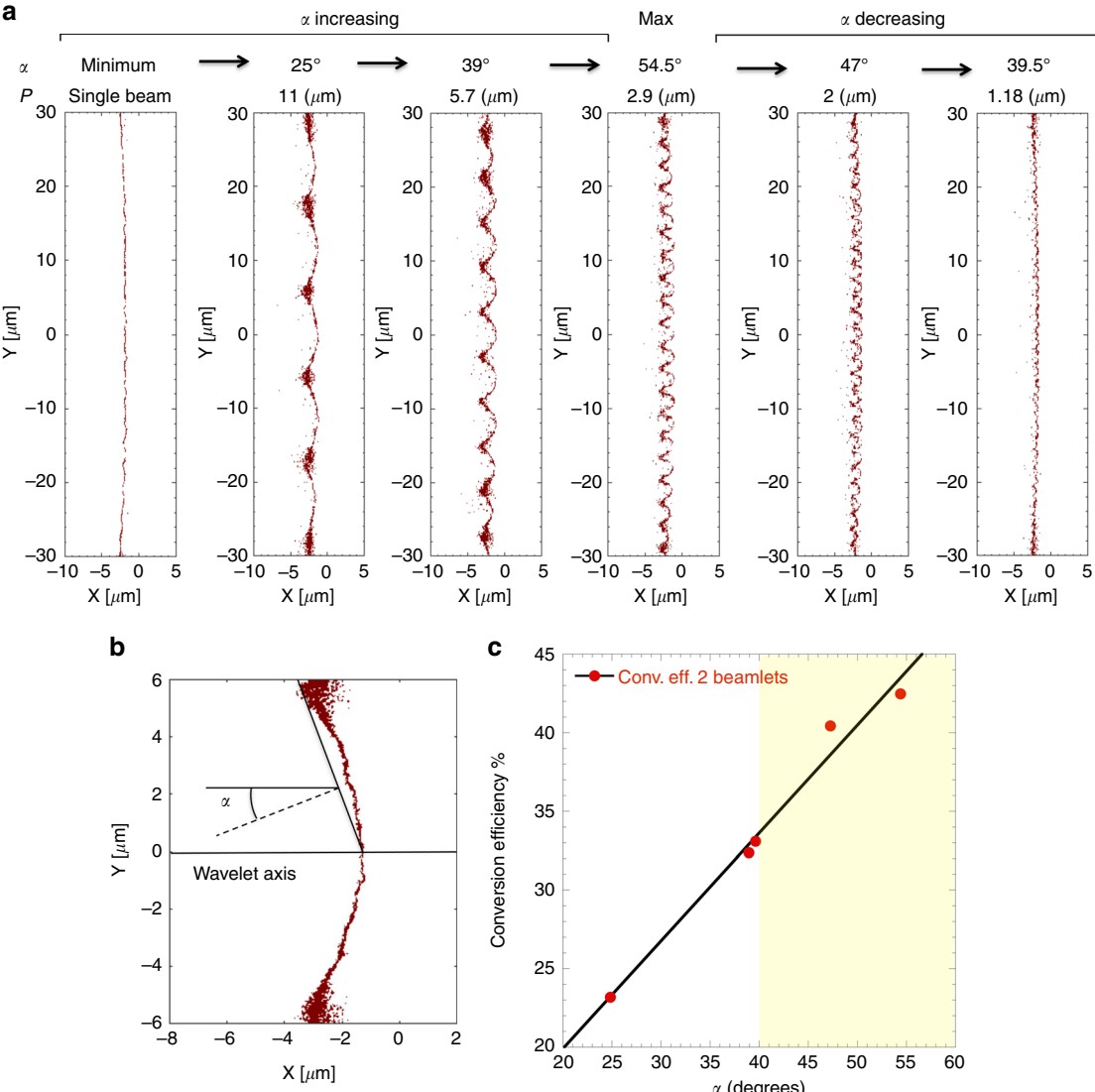

**Fig. 3** Definition of the modulation angle α and its relation to the conversion efficiency. **a** Plasma electron density profile with $1.5\,n_c \leq n_e \leq 6\,n_c$ for single beamlet and 2-beamlet interaction for all values of $P$ investigated. The single beamlet case shows a rather uniform density profile while the 2-beamlets cases shows a significant surface modulation with period corresponding to the interference intensity pattern. For larger periods ($2.9\,\mu m \leq P \leq 11\,\mu m$) the depth of the surface modulation is approximately constant, leading to larger values of α as P decreases. However, for $P < 2.9\,\mu m$ the modulation depth also decreases since the wavelets full width at half maximum approaches the electron excursion length in the laser field. This leads to the saturation of α reaching its maximum for $P = 2.9\,\mu m$ and subsequently decreasing for smaller values of P. **b** Representation of a single wavelet-induced modulation for $P = 11\,\mu m$. The average modulation angle α is defined as the average wavelet incidence angle on the plasma critical surface. **c** Laser-to-electron energy conversion efficiency as function of the average surface modulation angle α. The conversion efficiency linearly increases for all values of α. The dashed line-delimited area above 40 degrees represents the region where saturation and decline of conversion efficiency for forward moving electrons is expected

Fig. 4a, b are represented the $J_y$ component of the fast electron current density and the associated magnetic field for single wavelet-induced modulation and interference period $P = 11\,\mu m$. In presence of modulation the fast electron current, confined along the surface, flows towards the center where the associated magnetic field rapidly reverses sign, effectively injecting the fast electrons in the overdense plasma.

This mechanism can be directly observed by considering the cycle-averaged fast electron energy flux, confirming that a significant part of the fast electron energy is transported along the surface and injected into the overdense plasma at the modulation valley as shown in Fig. 4c, d. Therefore the magnetic field structure and amplitude associated to the surface modulation plays a fundamental role in the fast electron energy transport into the target material and then in the overall performance of

interfering beamlets LPI. A similar effect was described for much shorter pulses and curved targets in the modeling work by Ruhl and collaborators[32].

**Parametric study.** In the previous sections we exposed that interfering beamlets enhance laser energy coupling into fast electron and ion beams significantly compared to single beamlet interaction.

It is important to remark that performing simulations for interfering beamlets with s-polarized laser pulses, no significant difference is found between the single and two-beamlets interaction, as well as between different interference periods. This has two important implications to our work: on one hand it strengthens the thesis that vacuum heating is the major

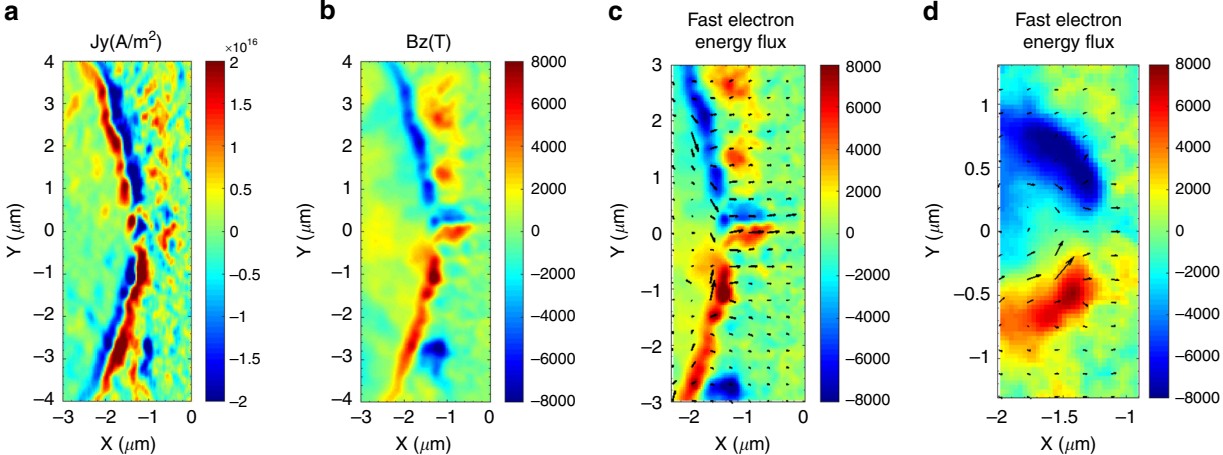

**Fig. 4** Current density and magnetic field maps. Current density and magnetic field maps restricted to a single wavelet interaction for $P = 11\,\mu m$ and $P = 2.9$ $\mu m$ and related fast electron energy flux. **a** Cycle-averaged y-component of the electron current density and **b** surface magnetic field structure at peak laser irradiation for $P = 11\,\mu m$. The fast electron current flows predominantly along the surface towards the center of the modulation/deformation and we can observe the corresponding return current flowing antiparallel to the fast electron current. A strong B-field structure develops between fast electron and return current effectively preventing the fast electrons from penetrating in the overdense plasma until they reach the modulation valley. Cycle-averaged fast electron energy flux (block arrows) superposed to the magnetic field snapshot at peak laser irradiation for **c** $P = 11\,\mu m$ and **d** $P = 2.9\,\mu m$. In both cases we observe that a large fraction of the fast electron energy flows along the surface confined by the balanced action of surface B-field and charge separation electric field and is finally injected in the overdense plasma at the modulation valley where the B-field rapidly reverses sign. This mechanism is responsible for the enhanced laser-to electron energy conversion efficiency even for large values of $\alpha$

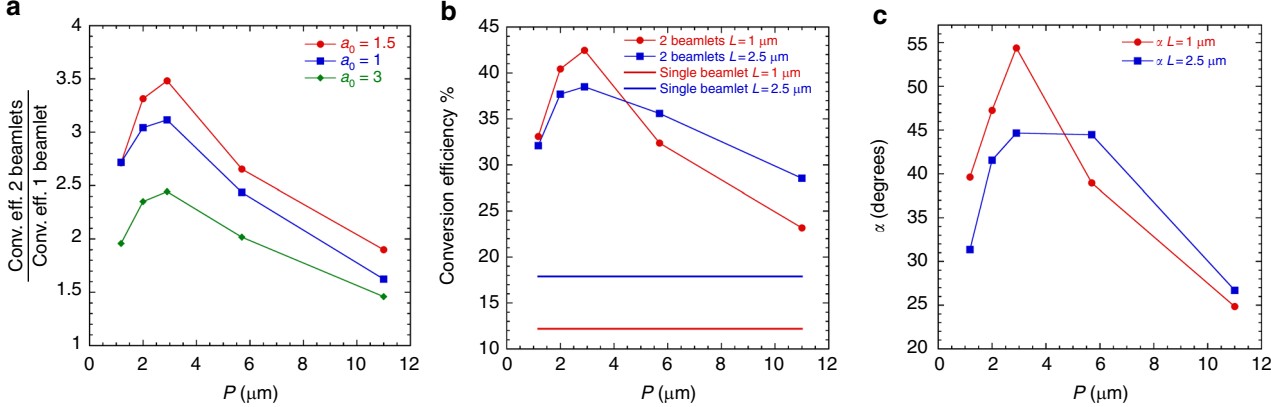

**Fig. 5** Parametric beamlet interference. Parametric study of beamlet interference for different laser intensities and pre-plasma scale-lenghts. **a** Conversion efficiency incremental factor as function of the interference period $P$ for laser intensities corresponding to $a_0 = 1$, $a_0 = 1.5$ and $a_0 = 3$. **b** Parametric study of laser-to-electron energy conversion efficiency for different pre-plasma scale-length $L$ as function of $P$. The red bulleted line and straight line refer to conversion efficiency for 2 and single beamlet irradiation and $L = 1\,\mu m$. The blue bulleted line and straight line refer to conversion efficiency for 2 and single beamlet irradiation and $L = 2.5\,\mu m$. As expected, higher conversion efficiency is found with longer pre-plasma scale-length for single beamlet interaction as well as multi-beamlet for large values of $P$. However, for $P < 5.9\,\mu m$ higher conversion efficiency is found for shorter pre-plasma scale-length and this relationship is well reproduced by **c** the trend of the average modulation angle $\alpha$ for the two scale-length cases, showing a similar trend compared to the conversion efficiency

mechanism responsible for laser energy absorption into hot electrons, on the other hand it also provides important information for laser and experimental design in case only two beamlets are available, since for two beamlets interaction the laser polarization must be parallel to the beamlets optical axes plane of incidence.

In this section we demonstrate that this behavior is robust and reproducible over a wide range of laser intensities and for longer pre-formed plasma scale-lengths. A set of simulations for interfering beamlets have been performed for $a_0 = 1$ and $a_0 = 3$, corresponding respectively to laser intensities of $1 \times 10^{18}$ and $1 \times 10^{19}\,W/cm^2$ and results are compared with single beamlet interaction of the same nominal intensity, keeping all the other simulation parameters constant as for the $a_0 = 1.5$ case. Figure 5a represents the conversion efficiency incremental factor for the three investigated intensities showing that the trend observed in the previous analysis is well reproduced for all laser intensities.

Finally we tested our approach for pre-formed plasma scale-length increased by 2.5 times, with pre-plasma extending 10 μm from the target surface and $a_0 = 1.5$. As expected, comparing the results with the previous set of simulations for the same laser intensity, the laser-to-electron energy conversion efficiency for single beamlet interaction is larger for longer pre-plasma scale-length and this behavior is maintained for two-beamlet interaction and large interference periods. However, for $P \leq 2.9\,\mu m$, although the conversion efficiency still remains significantly

higher than for single beamlet, the larger pre-plasma scale-length case underperforms compared to the shorter counterpart (Fig. 5b). This apparently unintuitive result can be explained once again looking at the trend of $\alpha$ for the two cases as shown in Fig. 5c, which closely reproduces the one for conversion efficiency, with modulation angle $\alpha$ for longer pre-plasma scale-length and $P \le 2.9$ μm smaller than the short scale-length case. This behavior is explained by considering the higher electron mobility in lower density plasmas, resulting in a faster closure of the wavelet-induced modulations.

In summary we find that the physics of intense interfering beamlets LPI can be broken down to three fundamental factors. The first is the wavelet self-focusing in the underdense plasma, that is especially important for large interference period and determines the temperature of the hot electron population. The second factor is the hole-boring induced modulation of the critical surface, that increases the wavelet incidence angle on target and boosts hot electron generation by vacuum heating-type mechanism. The third factor is the structure of the surface B-field, that increases the hot electron energy transport in the target material by injecting the surface-confined fast electrons into the overdense plasma independently from the modulation angle $\alpha$. This method can be applied for a wide range of laser intensities and laser contrast conditions on simple foil targets, proving its robustness compared to other methods implying specially designed structured targets and high-contrast femtosecond pulse lasers.

## Methods

**LFEX laser system**. The LFEX laser system is described thoroughly in ref. [7]. The four beamlets composing LFEX laser were focused on target by a single off-axis parabola with numerical aperture of F10 for each beamlet. The absence of final large deformable mirror optics (although 2 deformable mirrors per beamlet are placed at intermediate positions in the laser chain) limited the LFEX spot size to 60 μm, about double of the diffraction limited spot size.

No significant variation of spot size is observed between single and combined beamlets.

In an ideal case, focusing four identical and coherent beamlets, spatially and temporally overlapped results in two times reduction of the f-number, corresponding to a 4-fold increment in laser intensity. However, on LFEX as well as other large, multi-beamlet laser systems, such F-number reduction is strongly mitigated by the wave-front distortions of each single beamlet, as well as by wave-front distortion resulting from the combination of such beamlets.

The wave-front error in a large-scaled laser system could be classified into two kinds: static wave-front and dynamic wave-front error. The static wave-front error is related to the uniformity of the transmission optics and surface quality of both reflection and transmission optics. The dynamic wave-front error is affected by the main amplifier's thermal effect and nonlinearity (for either low energy or full-energy shots), the OPCPA nonlinearity and the stability of the optics. During a full-energy shot, only thermal effects in the amplifiers and non-linearities in the Nd: glass, whose contribution is limited, are added to the dynamic wave-front error. Therefore, it is expected that the interference pattern observed in the focal spot image shown in Fig. 1b is maintained during a full-energy shot.

In four-beamlet irradiation the total laser energy is equally distributed amongst the four beamlets with accuracy of about 5%. The amplified spontaneous emission prior to the main pulse has been measured to be negligible (amounting to few μJ) and LFEX main amplifiers are fired for all beamlets even for single beamlet operation. The only contribution to the laser pedestal is then constituted by the amplified optical parametric fluorescence from the front-end OPCPA, which amplitude is proportional to the total laser energy on target, therefore by keeping constant total laser energy we expect to have very similar pre-formed plasma conditions for single and multiple beamlet irradiation in the experiment. The LFEX contrast was recently improved by the addition of two saturable absorbers in the 3-stage OPCPA and slowly rises from $10^{10}$ at $t = -3$ ns to $10^9$ up to at $t = -150$ ps and exponentially increasing afterwards up to $10^4$ right before the main pulse arrival[33,34]. With a peak output intensity of $\sim 1 \times 10^{19}$ W/cm$^2$ this contrast level allows to successfully shoot thin foils (3–10 μm) generating bright TNSA ion sources.

**Particle in cell simulations**. Collisionless PIC simulations have been performed with the Epoch2d code using two different simulation setups focused on TNSA ion acceleration and fast electron generation. The choice for collisionless simulations resides in the fact that at relativistic or near-relativistic intensities the laser energy absorption mechanisms are eminently non-collisional. In fact, for quiver electron

energies exceeding few hundreds keV, the cross section for electron-ion collision at the critical density is significantly reduced, thus the contribution of collisional absorption mechanisms is negligible compared to non-collisional ones. The simulation box for ion acceleration was 200 μm in the longitudinal dimension and 150 μm in the transverse dimension with cell size λ/30 in both dimensions. The target has been modeled as a simple 5 μm Hydrogen foil with 50 particles per cell and 40 $n_c$ density with a sharp, 1μm scale-length exponential plasma profile, extending 5 μm from the front surface. The choice of Hydrogen instead of Aluminum is based on the fact that in relativistic LPI the laser energy absorption occurs through collisionless mechanisms, therefore no difference is expected between the two materials. Differences in the hole-boring velocity between the two materials do not affect the final surface modulation since the maximum depth for hole-boring is same[35].

The reduced laser energy used in these shots together with the high LFEX contrast justifies the choice of a sharp pre-plasma profile. Simulations for fast electron generation have similar plasma parameters in terms of material, density profile, particles per cell, and grid; however, the foil thickness have been increased to 15 μm and is attached to an absorbing boundary thus behaving as a semi-infinite plasma slab. A transverse extraction plane positioned 5 μm from the surface at 40 $n_c$ in the overdense plasma collected the information on fast electron momenta, allowing to obtain the energy spectra, angular divergence and laser-to-electron conversion efficiency.

## Data availability

The datasets analyzed during the current study are available from the corresponding author upon reasonable request. In accordance with the guideline for research data storage at the Institute of Laser Engineering, Osaka University, all data are properly stored in the SEDNA database system.

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

## Acknowledgements

The authors thankfully acknowledge the fruitful discussions with Dr. Zhaoiang Li on the properties of interfering beamlets on LFEX. The authors thank the technical support staff of ILE and the Cyber Media Center at Osaka University for assistance with the laser operation, target fabrication, plasma diagnostics, and computer simulations. This work was supported by the Collaboration Research Program between the National Institute for Fusion Science and the Institute of Laser Engineering at Osaka University, and by the Japanese Ministry of Education, Science, Sports, and Culture through Grants-in-Aid, KAKENHI (Grants No. 70724326, 15K21767, 15K17798, 17J02020, 15K21767, 15KK0163, 16K13918, 16H02245), Grants-in-Aid for Fellows by Japan Society for The Promotion of Science (Grant No. 14J06592, 14J06405, 15J00850, 17J02020, 17J07212, 18J11354) and Adaptable Seamless Technologytransfer Program through target driven R&D (A-STEP) of Japan Science and Technology Agency, "Development of key technologies for compact neutron source and its industrial application".

## Author contribution

A.M. designed and executed the experiment as principal investigator and performed the entire bulk of simulation work presented in the article. N.I., Y.S., and K.M. provided invaluable theoretical support. A.A. helped conceiving the experiment suggesting the investigation of LFEX beamlet interference effect. Y.A., A.Y., S.T., X.V., S.K., S.S., Y.Abe, S.L., K.M., N.K. and S.F. provided experimental support. T.N. prepared the targets used in the experiment, J.K., S.Tokita, and N.M. provided LFEX operation support. M.H., H.S., Y.Sakawa, M.N., H.N., H.A., S.F. and R.K. contributed to the discussion of the results. A.M wrote the paper with the contribution of N.I. Y.S. and S.F. The figures were prepared by A.M with the contribution of Y.Abe and N.I.
