## [Peer Review File · Nature Communications]

Reviewers' comments:

Reviewer #1 (Remarks to the Author):

The paper by Morace et al describes an experiment where an increased conversion efficiency from laser to hot electrons was observed. A 5 micrometer Aluminum (Al) foil was irradiated with 1 and 4 spatiotemporally coherent beams. The total energy approaching the target in both cases 270 J corresponding to a calculated intensity of approximately $2.5 \times 10^{18} \text{ W/cm}^2$.

The authors make the following claims:

- Overlapping multiple, coherent laser beams on target generates a mosaic-like interference pattern that will generate a critical surface modulation via radiation pressure and improve laser absorption compared to a single beam that would have equivalent laser energy. When the authors talk about a single beam they mean a uniform, unmodulated focal spot on target.
- Above hypothesis is experimentally supported by measurement of LPI-electrons and TNSA protons where they observe an increase of laser to electron conversion efficiency of 2.9x in the experiment with 4 beams and 2.7x in their model for 2 beams (for $E > 1 \text{ MeV}$).
- The individual beams break up in "wavelets" when interfering with each other at the target undergo self-focusing in the expanding plasma blow-off. This contributes to an increase of the effective laser intensity.

To support their claims authors developed a two-dimensional PIC simulation of two interfering laser beams interacting with a 5um hydrogen layer and a pre-formed plasma extending 5um and with scale length 1um.

Comments:

The observed enhancement in coupling efficiency is a very interesting and to the HED community important result. However, the physics explanation the authors offer in their paper are not convincing in my mind as there are many open questions. To name a few:

- 1) The experimental setup in fig 1 shows 4 beams that can be coherently overlapped on the target. There are two important coherence terms that play in space and time. First, four (ideal, i.e. perfect wavefront and temporal profile) beamlets that are adjacent to each other will lead to a reduction in $F\#$, therefore an intensity enhancement of approximately 4x for no gap, and even larger with more gap between beamlets. The gap though causes intensity modulation within the envelop of the diffraction limit of a single beam (Fourier theorem). Second, the authors assume for coherent combining only an increase of 2x in intensity and dropped the power (energy) in the experiment by the number of beams. However, this is only true for incoherent beams; coherent beams will see a 4x increase in intensity in their maxima.
- 2) The intensity pattern shown from the frontend shows strong a strong speckle pattern due to beam imperfections and coherent beam interference. It is unclear:
 - a. Whether the observed pattern is similar and/or maintained for a full system shot
 - b. What the image depth of each speckle pattern is, i.e. how the speckle pattern evolves in beam propagation direction. Due to the extreme high spatial frequencies these speckle patterns evolve quickly over short distances of the order of several wavelengths. Authors may need to evaluate the impact of this effect in correspondence to their hypothesis of self-focusing in the under-dense plasma (the effects might wash-out).
 - c. What the temporal coherence function of each beamlet is, and whether the interference pattern is static during overlap or whether it evolves with pulse duration.
- 3) Details are missing for why specific initial conditions are chosen for the simulations, e.g. how was the preformed plasma chosen, why a hydrogen target was modeled (vs Aluminum in experiment) and/or why this approximation is valid.
- 4) Simulations were conducted with only 2 beams interfering, and power/intensity levels are inconsistent with coherent overlap of beams and boost in $F\#$ as explained before. Authors should also state why 2 vs 4 beams model is adequate and/or what differences should be expected.

- 5) Reason should be given for the chosen cell size and how it corresponds with the interference pattern period. Was convergence demonstrated?
- 6) A conclusion is drawn that laser contrast doesn't play a role. No reason is given how the authors arrived at this claim, and what lower or upper boundaries for temporal contrast would be.
- 7) Finally, the summary calls out several mechanisms for the enhancement of the particle characteristics in the multi-beam interaction. None of these are actually demonstrated in either the simulations or the presented experimental data.

Reviewer #2 (Remarks to the Author):

The paper by Morace et al. presents a study of laser-solid target interaction in a very special regime when the laser spatial profile is modulated due to interference of multiple beamlets. The authors argue that in this regime energy conversion from laser pulse to hot electrons is enhanced due to a number of reasons, and this method has advantages over conventional micro/nano modifications of the target surface (namely, the new method is suitable for low-contrast laser pulses). The physics of the new phenomena is explored mainly through PIC-simulations.

The simulation section of the paper is very elaborate and is complemented with thorough discussions of the underlying physics. At the same time, I found the experimental section unproportionally weak. The only experimental data presented are 4 curves – the spectra of hot electrons and protons for 1 and 4 beamlets cases. These data correspond to maximum of 4 experimental shots (or even 2). There is no information on stability of the results or any error analysis. The unbalance between the simulations and experiment significantly weakens the authors' claims to a point where one can question their validity. For example, will the energy conversion enhancement persist for lower-contrast laser beams? Is there an optimum for the interaction angle of the beamlets? While the simulations provide the answers, they are not backed up by the experimental data.

On a side note, the authors might broaden the introduction by mentioning an alternative approach to create a spatially modulated plasma density profile on the target surface, as discussed in "Optically Controlled Solid-Density Transient Plasma Gratings", Monchocé et al., Phys. Rev. Lett. 112, 145008 (2014). Also, please look into Figure 3, it seems damaged (especially the bottom-right panel). Finally, I can't recommend the paper for publication in its current form. The authors should provide more comprehensive experimental data and tie it with the simulations to balance the paper and strengthen their claims.

The paper by Morace et al describes an experiment where an increased conversion efficiency from laser to hot electrons was observed. A 5 micrometer Aluminum (Al) foil was irradiated with 1 and 4 spatiotemporally coherent beams. The total energy approaching the target in both cases 270 J corresponding to a calculated intensity of approximately $2.5 \times 10^{18} \text{ W/cm}^2$.

The authors make the following claims:

- Overlapping multiple, coherent laser beams on target generates a mosaic-like interference pattern that will generate a critical surface modulation via radiation pressure and improve laser absorption compared to a single beam that would have equivalent laser energy. When the authors talk about a single beam they mean a uniform, unmodulated focal spot on target.*
- Above hypothesis is experimentally supported by measurement of LPI-electrons and TNSA protons where they observe an increase of laser to electron conversion efficiency of 2.9x in the experiment with 4 beams and 2.7x in their model for 2 beams (for $E > 1 \text{ MeV}$).*
- The individual beams break up in “wavelets” when interfering with each other at the target undergo self-focusing in the expanding plasma blow-off. This contributes to an increase of the effective laser intensity.*

To support their claims authors developed a two-dimensional PIC simulation of two interfering laser beams interacting with a 5um hydrogen layer and a pre-formed plasma extending 5um and with scale length 1um.

Comments:

The observed enhancement in coupling efficiency is a very interesting and to the HED community important result. However, the physics explanation the authors offer in their paper are not convincing in my mind as there are many open questions. To name a few:

1) The experimental setup in fig 1 shows 4 beams that can be coherently overlapped on the target. There are two important coherence terms that play in space and time. First, four (ideal, i.e. perfect wavefront and temporal profile) beamlets that are adjacent to each other will lead to a reduction in F#, therefore an intensity enhancement of approximately 4x for no gap, and even larger with more gap between beamlets. The gap though causes intensity modulation within the envelop of the diffraction limit of a single beam (Fourier theorem).

The beamlets are coherent but not phase-matched and therefore undergo interference in the focal spot.

In order to better expose the LFEX laser system we show a schematic of LFEX laser. The LFEX system is composed by a single oscillator and OPCPA front end. The beamlets are separated quite late in the laser chain, after the first rod amplifier when the total laser energy is about 10 J. After that the laser beamlets move on independent, but virtually identical paths up to TCC (note that DM stays for deformable mirror and OAP for off-axis parabola).

Figure 1: Schematic of LFEX laser system

If the beamlets were to be phase-matched then the $F\#$ would be reduced for 4 beamlets irradiation as the referee points out, but no interference pattern would occur since the overlapping would result in a single laser beam. In order to achieve full phase matching between the beams, four large (50 cm x 50 cm) deformable mirrors are required, together with an automatic feedback control system to adjust the optical path length and compensate for the micrometer scale optical path fluctuations along the LFEX chain due to thermal effects.

For sake of clarity this consideration has been added in the methods section with the following sentences:

“The absence of final deformable mirror optics as well as of an automatic feedback control system for optical path adjustment also implies that the four beamlets cannot be phase-matched and therefore do not constitute a single laser beam when overlapped spatially and temporally. Therefore the four beamlets have to be considered as coherent (as they are split earlier in the LFEX chain from a single pulse) but independent, therefore maintaining the same numerical aperture (or $F\#$) as the single beamlet when overlapped.”

Second, the authors assume for coherent combining only an increase of 2x in intensity and dropped the power (energy) in the experiment by the number of beams. However, this is only true for incoherent beams; coherent beams will see a 4x increase in intensity in their maxima.

The referee is correct. Four coherent beams with identical spatial and temporal profile will see a 4x increase in laser intensity in their maxima.

Nevertheless, the goal of the experiment and the paper itself is to demonstrate that given constant total laser energy, pulse duration and focal spot size, interfering beamlets guarantee much higher laser beam performances compared to the single beam. In this sense the increment of laser intensity in the maxima compared to the nominal (average) intensity on target is another argument in favor of using multiple beamlets.

We must remark that in our experimental conditions the four beamlets slightly differ in terms of spatial intensity profile, focusing and temporal compression. Therefore we do not expect a perfect 4x increment.

In any case to clarify we added modified the abstract, specifying that we refer to “nominal” intensity on target.

2) The intensity pattern shown from the frontend shows strong a strong speckle pattern due to beam imperfections and coherent beam interference. It is unclear:

a. Whether the observed pattern is similar and/or maintained for a full system shot

The pattern is recorded not in the front-end but in the laser focal spot in target chamber center (TCC). The beamlets therefore travel the full system path, are compressed to 1.5 ps and focused by the off-axis parabola. The image is obtained using an equivalent plane monitor imaging the laser focal spot focused by the off-axis parabola. We specified in the Figure 1b caption that the focal spot image is recorded in TCC.

b. What the image depth of each speckle pattern is, i.e. how the speckle pattern evolves in beam propagation direction. Due to the extreme high spatial frequencies these speckle patterns evolve quickly over short distances of the order of several wavelengths. Authors may need to evaluate the impact of this effect in correspondence to their hypothesis of self-focusing in the under-dense plasma (the effects might wash-out).

We thank the referee for the remark.

The depth of focus for LFEX beamlets is calculated to be 240 μm . Therefore we have clear interference within 240 μm along the focal axis and it is much larger than the laser-plasma interaction region.

Moreover the focal spot image in Figure 1 is time integrated (being 1.5 ps pulse duration and recorded by a 12 bit CCD camera), therefore it represents the entirety of the laser pulse, the speckles (or interference) maxima are therefore maintained throughout the laser irradiation.

We also modified Figure 1b for more clarity.

The added sentence to methods section is:

“It is important to remark that the beamlets coherence throughout the pulse duration is demonstrated by the focal spot image acquired in TCC and represented in Fig. 1b. The image is time-integrated, thus confirming that the interference pattern is maintained over the pulse duration. Such equivalent plane monitor images are routinely acquired on LFEX laser and display interference patterns whenever the beamlets are spatially and temporally overlapped.”

c. What the temporal coherence function of each beamlet is, and whether the interference pattern is static during overlap or whether it evolves with pulse duration.

As we mentioned in the reply to question 2b, the image recorded in Figure 1 is the actual time-integrated focal spot profile recorded in TCC. Therefore the interference pattern is static during the pulse duration.

3) Details are missing for why specific initial conditions are chosen for the simulations, e.g. how was the preformed plasma chosen, why a hydrogen target was modeled (vs Aluminum in experiment) and/or why this approximation is valid.

The pre-formed plasma scale-length was chosen based on previous measurements performed on LFEX, that is possible to find in references 31-32. The LFEX contrast is naturally very high, allowing to shoot micrometer size targets even at full energy shots. In our experimental conditions we are shooting at about 25 % of the LFEX full energy and therefore the pre-plasma scale length is expected to be very short, at the micrometer level, being the pedestal energy proportional to the pulse energy.

As for the choice of using hydrogen instead of Aluminum, we remark that relativistic intensity laser energy absorption occurs through collisionless mechanisms so that no difference appears between Hydrogen and Aluminium. This is also the reason why we performed collisionless 2D PIC simulations.

Although the hole boring velocity is different between Hydrogen and Aluminum (given the same electron density in the simulation), the maximum depth for hole boring is same, as we demonstrated in Iwata et al. Nature Communications 9, 623 (2018) that we added to the references.

In our conditions the final surface modulation, which determines the laser energy absorption, is expected to be same between Al and H.

Since 2D simulations cannot fully reproduce experimental data in any case (3D are required), we decided to use simple Hydrogen ions cause this is sufficient to explain and clarify the underlying physics.

“The choice of Hydrogen instead of Aluminum is based on the fact that in relativistic LPI the laser energy absorption occurs through collisionless mechanisms, therefore no difference is expected between the two materials. This is also the reason why collisionless PIC simulations were chosen. Differences in the hole-boring velocity between the two materials do not affect the final surface modulation since the maximum depth for hole boring is same [Iwata et al]”.

4) Simulations were conducted with only 2 beams interfering, and power/intensity levels are inconsistent with coherent overlap of beams and boost in F# as explained before. Authors should also state why 2 vs 4 beams model is adequate and/or what differences should be expected.

Given that 3D simulations are intrinsically more accurate than 2D simulations, we think that the representation of the Physics is actually quite accurate in the 2D case given the following reason: the most of the laser energy absorption occurs in the laser polarization plane since the leading mechanism is vacuum heating.

To verify this and rule out other phenomena, as for example the increment in the interaction area due to surface modulation, we performed 2D simulations with s-polarized laser light, which shown substantially no effect of beamlet interference on laser energy absorption. Therefore the laser energy absorption for interfering beamlets occurs in an almost bi-dimensional geometry being determined by the polarization plane.

This is discussed in the parametric study section of the manuscript.

5) Reason should be given for the chosen cell size and how it corresponds with the interference pattern period. Was convergence demonstrated?

The laser wavelength needs to be sampled with many points in order to obtain accurate LPI simulations. In this work we chose the cell size to be $\lambda/30$ as mentioned in the methods section, which is also about $1/35$ of the minimum interference period ($1.18 \mu\text{m}$). We consider this resolution to be sufficiently high to produce results as accurate as higher resolutions would.

6) A conclusion is drawn that laser contrast doesn't play a role. No reason is given how the authors arrived at this claim, and what lower or upper boundaries for temporal contrast would be.

In the paper we said that the interference effect is rather insensitive to a wide range of laser contrast conditions. The contrast of LFEX laser is already very high as reported in references 31-32. Especially for low energy shots, as in our experimental conditions, the pre-plasma scale length is expected to be very low. We performed simulations in complete absence of pre-plasma still showing increment in the conversion efficiency as reported in figure 2. However to experimentally verify this, a plasma mirror should be implemented on LFEX laser.

Figure 2, fast electron spectra obtained in complete absence of pre-plasma, the blue curve represents the electron spectrum for single beamlet and the red curve the spectrum for interfering beamlets.

Moreover we verified, in the parametric study session (figure 5b in the manuscript), that by increasing the pre-plasma scale-length and spatial extension by more than a factor 2 the interference effect is still significant. The lower boundary for laser contrast corresponds to the case where the pre-plasma scale-length is so long that the single beamlet would undergo filamentary instability in the pre-formed plasma, in fact resulting in a structure similar to interference pattern.

7) Finally, the summary calls out several mechanisms for the enhancement of the particle characteristics in the multi-beam interaction. None of these are actually demonstrated in either the simulations of the presented experimental data.

The 3 mechanisms at the base of the laser energy absorption enhancement for interfering beamlets have been clearly individuated in our simulation work and exposed in dedicated sections of the manuscript: i) Wavelet self-focusing, ii) Surface modulation, iii) Surface magnetic field.

These three mechanisms combined lead to higher laser energy absorption into hot electrons as it is demonstrated in the simulations.

Reviewer #2 (Remarks to the Author):

The paper by Morace et al. presents a study of laser-solid target interaction in a very special regime when the laser spatial profile is modulated due to interference of multiple beamlets. The authors argue that in this regime energy conversion from laser pulse to hot electrons is enhanced due to a number of reasons, and this method has advantages over conventional micro/nano modifications of the target surface (namely, the new method is suitable for low-contrast laser pulses). The physics of the new phenomena is explored mainly through PIC-simulations.

The simulation section of the paper is very elaborate and is complemented with thorough discussions of the underlying physics. At the same time, I found the experimental section unproportionally weak. The only experimental data presented are 4 curves – the spectra of hot electrons and protons for 1 and 4 beamlets cases. These data correspond to maximum of 4 experimental shots (or even 2). There is no information on stability of the results or any error analysis. The unbalance between the simulations and experiment significantly weakens the authors' claims to a point where one can question their validity data.

We thank the referee for carefully evaluating this paper and for his comments.

Performing experiments on large laser facilities such as LFEX, NIF and NIF-ARC necessarily results in a reduced number of shots that can be performed. Experimental campaigns are composed by 2-4 laser shots and new experimental proposals investigating the physics of previously performed experiments are usually not accepted. Although the lack of statistics may appear (and in some cases is) as a limitation to the scientific results, we are very confident about the robustness of our experimental results for the following reasons.

Typical shot-to-shot fluctuations in terms of energy/intensity on LFEX laser amount to 10-15% difference in laser energy absorption into hot electrons. In our experimental results we obtain a 200% improvement in laser-to-electron energy conversion efficiency and 220% improvement in laser-to-proton energy conversion efficiency, with same total laser energy and nominal intensity on target (energy fluctuation of only 7%). This unprecedented result cannot be explained in any other way than concluding that the beamlet interference plays a key role in enhancing the laser-to-electron energy conversion efficiency and electron temperature.

The reason why this experiment was conducted resides in the observation of extremely bright and energetic proton beams obtained via full energy shots with LFEX laser on 10 μm Al foils.

At full energy, LFEX laser delivers up to 1 kJ of laser energy on target with $a_0 = 2.7$ (intensity $\sim 1 \times 10^{19}$ W/cm²). Results from previous experiments show that by irradiating 10 μ m Al foils at full energy with beamlets temporally and spatially overlapped, we obtained proton beams with 53 MeV maximum energy and laser-to-proton energy conversion efficiency of $\sim 8\%$ for protons energies exceeding 3 MeV. These results have been obtained in a reproducible way on multiple shots and recorded by Thomson Parabola as well as radio-chromic film stack diagnostic, that independently confirmed the reproducibility of the data (see the figure below).

Such high peak energies and conversion efficiencies are far above those predicted by existing scalings, and therefore led us to formulate the hypothesis that the beamlet interference played an important role for laser energy absorption into energetic particle beams.

On a side note and as confidential information, this work was presented as invited talk in several international conferences, and the PI has recently being contacted from CEA's LMJ-Petal team in France, where apparently extremely high proton cut-off energies and high conversion efficiencies using the Petal laser system have been observed. The results are similar to the LFEX performances and above any existing scaling. The Petal laser system is composed by 4 beamlets focused by a single parabola, and our common opinion is that interference plays an important role on that laser system too.

Figure 1 a) Thomson Parabola spectra of proton beam generated by irradiating 10 μ m Al foils at full LFEX energy (~ 1 kJ on target). b) Radio-chromic film data obtained by a different shot in conditions similar to a), the two diagnostics independently confirm the experimental data.

For example, will the energy conversion enhancement persist for lower-contrast laser beams?

In the parametric study section of our manuscript we verified that laser energy absorption into energetic electrons is enhanced for interfering beamlets also in case of larger pre-plasma scale length (see Fig. 5b in the manuscript), corresponding to lower contrast conditions.

The enhancement will persist until the pre-plasma scale-length is so long that the single beamlet would undergo filamentary instability in the pre-formed plasma, in fact resulting in a structure similar to interference pattern.

Is there an optimum for the interaction angle of the beamlets? While the simulations provide the answers, they are not backed up by the experimental data.

Since the LFEX laser beamlets are focused using the same off-axis parabola, it is not possible to experimentally study the presence of an optimum incidence angle for the beamlets. The only way to change the beamlets incidence angle is to use a focusing plasma mirror such as an ellipsoidal or parabolic plasma mirror thus changing the beam F#. However we thoroughly investigated the dependence of laser-to-electron energy conversion efficiency on the beamlets incidence angle in the manuscript by simulation (see Table 1 and Figures 2-3).

On a side note, the authors might broaden the introduction by mentioning an alternative approach to create a spatially modulated plasma density profile on the target surface, as discussed in “Optically Controlled Solid-Density Transient Plasma Gratings”, Monchocé et al., Phys. Rev. Lett. 112, 145008 (2014). Also, please look into Figure 3, it seems damaged (especially the bottom-right panel).

We thank the referee for the suggestion and we added this paper as reference and mentioned this work in the introduction by adding the sentence :

“Recently the generation of transient plasma gratings using 2 interfering pre-pulses was demonstrated²⁸, with several potential applications to LPI especially with ultra-short Ti:Sa laser systems.”

We will verify if there is any corruption in the pdf file generated upon submission.

Finally, I can't recommend the paper for publication in it's current form. The authors should provide more comprehensive experimental data and tie it with the simulations to balance the paper and strengthen their claims.

Although the referee concern is understandable, we disagree with his conclusion.

As we already discussed above, experiments on kJ-class laser facilities are necessarily constituted by few laser shots. Nevertheless the experimentally observed

extraordinary enhancement in laser-to electron and proton energy conversion efficiency is not in any way attributable to statistic fluctuations. Moreover the experimental results are in good agreement with the simulations for beamlet interference (see Fig 1 c-f) with parameters corresponding to the experimental conditions.

Based on the previous discussion we kindly ask the referee to reconsider the manuscript.

Reviewers' comments:

Reviewer #1 (Remarks to the Author):

I appreciate the response by the authors and for considering my comments. In my opinion, further consideration is required to underpin the observations with a conceivable physics explanation. Specifically, in order that this manuscript be worthy of Nature Communications, authors should further review the concept of optical coherence and its impact for this paper. Specifically, when overlapping beams, consideration should be given whether the local, momentum E-field or the pulse envelope drives the observed physics.

(1) My first concern addressed the coherence in space and time and the intensity enhancement through an effective increase in the F number. The Authors' response isn't fully clear as they distinguish between phase matching and interference. Phase matching is typically a term used in non-linear optics where incoherent beams are phase matched and coherently coupled through a nonlinear electronic response of a dielectric medium. The authors argue that the effective F number increase must not be considered but the intensity pattern in the focus resulting from beam interference should be considered. This explanation is inconsistent and incorrect.

The F number effect will only not play a role if the beams originate from the same, single beam aperture.

Beams are either coherent and so there is an effective change in F number and the beams generate an intensity pattern at focus with a static intensity distribution over the pulse length (more or less), or the beams are incoherent, in which case there is a rapidly changing intensity pattern at focus – impossible to resolve with conventional diagnostics. What matters is the effective coherence length laterally and temporally.

In their rebuttal, the Authors say "Nevertheless, the goal of the experiment and the paper itself is to demonstrate that given constant total laser energy, pulse duration and focal spot size, interfering beamlets guarantee much higher laser beam performances compared to the single beam. In this sense the increment of laser intensity in the maxima compared to the nominal (average) intensity on target is another argument in favor of using multiple beamlets."

I very much appreciate the observation, purpose and intent of the experiment; however, the observations must be accompanied by a current and compelling physics explanation.

(2) Coherence, continued

a. The Authors' response indicate that the intensity pattern is recorded in a conjugated plane.

However, it remains unclear what the spot pattern is during the experiment in a full system shot. This should be clarified in the manuscript.

b. Authors say that the inference pattern is static throughout the focus and the pulse duration. This is only true if the beams are quasi-plane wave; the appearance of interference effectively shortens the focal depth of the beam – again because of a reduced coherence length in time due to beam angle and distortions.

c. A time-integrated measurement that yields a "frozen" pattern does not conclude a static interference pattern over the spectrum of intensities the target is exposed to.

(3) The given explanation is still unsatisfactory. The initial choice of parameters in the simulations still need to be explained in the manuscript. Second, I frankly cannot follow the other explanation, even after studying the Iwata reference. Authors added: "[...] is based on the fact that in relativistic LPI the laser energy absorption occurs through collisionless mechanisms, therefore no difference is expected between the two materials. This is also the reason why collisionless PIC simulations were chosen. [...]" Authors should explain, what the collisionless mechanisms are, why velocity throughout the layer doesn't matter, and why it can be treated solely relativistic when the laser itself is just bordering on relativistic (2.5×10^{18} W/cm²) and undoubtedly is comprised of a profile of intensities which dips below relativistic at the outer edges of the beam.

(4) I think this point needs to be proven by example, particularly since the response to #3 states that 2D cannot reproduce experimental data. Also, the polarization of the laser(s) used in the experiment are not discussed in the manuscript – if they were S-polarized, this should be stated, along with the

explanation written by the Authors in the rebuttal.

(5) Was convergence demonstrated?

(6) It is recommended to only describe the observation from the parametric study but not to make a claim that contrast doesn't play a role. Maybe a threshold could be estimated along with a corresponding time length.

(7) Three explanations are offered. Wavelet self-focusing: This needs to be tied back to the coherence functions (including 2 vs 4 beam simulations). Surface modulation and Magnetic fields at the surface: Table (1) summarizes the parameter study ranging from 2.6 to 25 degrees. Experimental observations were made at the smallest angle, 2.6 degrees. If I understand Fig 3 correctly, the surface magnetic field would not play a role in the experimental configuration. Authors should tie simulations/explanations more closely to the experimental observations to be consistent with the journal's guidelines.

Reviewer #2 (Remarks to the Author):

My main concern with the paper was the lack of experimental data. While the authors did not add any new data to the paper after the revision, they argued that the results they presented are reliable since the effect they observed is much stronger than what would be expected based on the level of fluctuations of laser energy and other experimental parameters. I think this is a valid claim. I also understand that it is challenging to take significant amount of shots on a kJ-level laser system, even though I am certain that an experimental run can not result in only two shots (demonstrated in the paper), so the authors should have more data.

Nevertheless, I believe that the paper is important for the high-power laser community. The simulations section is very elaborate and does a great job explaining the experimental results. For these reasons I can recommend the paper for publication.

Reviewer #3 (Remarks to the Author):

This review is in addition to the previous two reviewers' review and their responses.

As reflected by other reviewers and the authors themselves, the manuscript presents experimental results and simulation-supported analysis of their results to derive claims that are particularly interesting to the HED community. More specifically, the authors present a significant efficiency enhancement in electron and ion conversion from the LFEX system when using angled non-coherent beamlets (compared to the same nominal energy); and identify the key players in this enhancement, namely beamlet self-focusing and converging surface magnetic fields, employing reduced dimensionality PIC simulations.

I believe that the authors have addressed relatively well the concerns raised by Reviewer 1 and 2, which were thoroughly critical in questioning the validity of the presented experimental results and conclusions. The authors have accommodated the answers to the concerns raised and have improved the overall quality of the original manuscript.

I would still like to echo the most general concerns, which are not straight-forward to address in practice, with regards to the relatively limited experimental data supporting the conclusions reached in the manuscript. While the measured ion conversion efficiency enhancement is unarguable from the experimental data alone, a more comprehensive set of experimental results would shine more light onto an otherwise fine set of simulations and resulting conclusions. Ultimately, the simulations should be used to complete the circle and validate in some of the details concerning the experimental parameters and conclusions. However, in the present form of the manuscript, the experimental results have a number of shortcomings and simulations are used to fill in relatively large voids, thus risking

misinterpretation of the results. Examples of these experimental shortcomings are the lack of beamlet angle comparative results, a better characterized composite laser beam, and a more comprehensive 4-beamlet simulation.

On the other hand, one must understand the complexity of systems like LFEX and consider valid explanations for some of these shortcomings offered by the authors in their responses to reviewers. Requiring to address all raised concerns in this case, particularly in the case where they may not be practically feasible, while of utmost importance, may hamper progress and dissemination of otherwise high-quality high-impact results.

In light of these comments, I believe the manuscript is of sufficient overall quality to be considered for Nat. Comms., although it would be of unambiguous impact with supporting experimental results.

Reviewer #1 (Remarks to the Author):

I appreciate the response by the authors and for considering my comments. In my opinion, further consideration is required to underpin the observations with a conceivable physics explanation. Specifically, in order that this manuscript be worthy of Nature Communications, authors should further review the concept of optical coherence and its impact for this paper. Specifically, when overlapping beams, consideration should be given whether the local, momentum E-field or the pulse envelope drives the observed physics.

(1) My first concern addressed the coherence in space and time and the intensity enhancement through an effective increase in the F number. The Authors' response isn't fully clear as they distinguish between phase matching and interference. Phase matching is typically a term used in non-linear optics where incoherent beams are phase matched and coherently coupled through a nonlinear electronic response of a dielectric medium. The authors argue that the effective F number increase must not be considered but the intensity pattern in the focus resulting from beam interference should be considered. This explanation is inconsistent and incorrect. The F number effect will only not play a role if the beams originate from the same, single beam aperture.

Beams are either coherent and so there is an effective change in F number and the beams generate an intensity pattern at focus with a static intensity distribution over the pulse length (more or less), or the beams are incoherent, in which case there is a rapidly changing intensity pattern at focus – impossible to resolve with conventional diagnostics. What matters is the effective coherence length laterally and temporally. In their rebuttal, the Authors say “Nevertheless, the goal of the experiment and the paper itself is to demonstrate that given constant total laser energy, pulse duration and focal spot size, interfering beamlets guarantee much higher laser beam performances compared to the single beam. In this sense the increment of laser intensity in the maxima compared to the nominal (average) intensity on target is another argument in favor of using multiple beamlets.”

I very much appreciate the observation, purpose and intent of the experiment; however, the observations must be accompanied by a current and compelling physics explanation.

We apologize to the reviewer for using improper terminology, we are aware of phase-matching in non-linear optics and it is not what we meant with that, we intended instead that the beams are not overlapped with the same exact phase.

The reviewer is correct, in an ideal case four coherent beamlets, spatially and temporally overlapped with same phase and identical wave-front will result in a single focal spot with reduction of F-number. Precisely if the beam diameter is doubled then the F-number is expected to be reduced by half.

However in our experiment, since the beam aperture is very large and the laser system is also very complicated, each beamlet possesses a significant wave-front distortion, even though deformable mirrors are used in the initial stages of amplification for each beamlet. Meanwhile, during the coherent combination (from four beamlets to a single combined beam), because of the absence of a big aperture deformable mirror, four

beamlets actually are not exactly in phase. In this case, the combined beam possesses a seriously distorted wave-front, which seriously degrades the focal spot. In theory, apart from the wavelength, the quality of a focal spot is mainly determined by the *F number* and the wave-front. And, unfortunately, the degradation sensitivity of a focal spot to its wave-front increases with reducing the F-number. Therefore although a reduction of F-number is expected, this does not necessarily translate in a significant focal spot reduction due to the highly degraded wave-front. It is extremely difficult though to provide quantitative information about this effect. We have modified the manuscript including these considerations in the methods section and we include here in the reply a typical image of the LFEX focal spots before and after overlapping, taken at the time of the experiment. As it is possible to notice the overall focal spot size does not vary (or it is even larger) for overlapped beamlets.

In particular in the experiment, the single beamlet shot was performed using the H1 beamlet, chosen because of the better focusing/ higher intensity compared to the other 3 beamlets. The H1 beamlet shows better focusing compared to the 4 beamlets combined.

Also the reviewer is correct in assuming that in an ideal case, interference of four coherent beamlets will result in 4x increment in the peak laser intensity in the interference maxima.

However the increased laser energy absorption into hot electrons observed in this work cannot be addressed by this effect, since it shows a clear dependence on the modulation angle α .

At the same time, the 4-fold increment (or 2-fold in simulation) in peak laser intensity is constant for all interference cases, which in absence of self-focusing should lead to similar hot electron temperatures. However in the simulations appear clear that the main contribution to the hot electron temperature (not to the conversion efficiency) is given by the wavelet self-focusing (see fig. 2d).

(2) Coherence, continued

a. The Authors' response indicate that the intensity pattern is recorded in a conjugated plane. However, it remains unclear what the spot pattern is during the experiment in a full system shot. This should be clarified in the manuscript.

The reviewer's comment is quite correct. The wave-front error in a large-scaled laser could be classified into two kinds: static wave-front and dynamic wave-front. The static wave-front is related with uniformity of transmission optics and surface quality of both reflection and transmission optics. The dynamic wave-front is affected by the main amplifier's thermal effect and nonlinearity (with and without full energy), the OPCPA nonlinearity, the stability of optics, etc. In our measurement, the static and major dynamic wave-front errors are considered, and only the full energy induced dynamic wave-front (thermal effect and nonlinearity in Nd:glass) is not included. According to our previous measurement, compared with above factors, it is not a major one and whose influence here is very limited.

Below figure is Fig. 7 from the reference of (J. Bromage, S.-W. Bahk, D. Irwin, J. Kwiatkowski, A. Pruyne, M. Millecchia, M. Moore, and J. D. Zuegel, "A focal-spot diagnostic for on-shot characterization of high-energy petawatt lasers," *Opt. Express* 16, 16561-16572 (2008)), which shows the focal spot of the omega-EP PW laser at LLE, the University of Rochester, a Nd:glass high-energy petawatt laser and almost same with our laser. (a) is the full energy focal spot, (b) is the low energy focal spot (Nd:glass amplifier is not working), and (c) is the ideal diffraction limit focal spot. From (a) and (b), we can find that the full energy thermal effect and nonlinearity only slightly changes some details, but the overall distribution is not changed. And from (a)(b) and (c), the actual focal spots (a)(b) deviate far away from the ideal one (c) due to wave-front errors. It is why a quarter focal spot area and a 4 times enhanced intensity cannot be obtained.

b. Authors say that the inference pattern is static throughout the focus and the pulse duration. This is only true if the beams are quasi-plane wave; the appearance of interference effectively shortens the focal depth of the beam – again because of a reduced coherence length in time due to beam angle and distortions.

The image depth of each speckle pattern (or each beamlet) is around 240 mm. Actually the weight of high spatial frequencies in each beamlet is not very high. The messy distributed focal spot is due to the interference of four distorted focal spots from four segment beamlets. Because the image depth is 240 μm , the evolution of the

coherently overlapped focal spot is not very sensitive to the propagation direction. Below is the Fig. 12 also from the above reference, which shows the focal spots at different propagation locations. Even if the detection plane is shifted with a length of $\sim 200 \mu\text{m}$, the spot has a similar distribution.

c. A time-integrated measurement that yields a “frozen” pattern does not conclude a static interference pattern over the spectrum of intensities the target is exposed to.

Our pulse possesses a near Gaussian profile with a duration of around 1.5 ps. The spectral bandwidth is very narrow around 3 nm (FWHM), which almost can be considered as a near monochromatic wave. In this case, the temporal evolution is very limited, and the spatially distorted focal spot can be considered to be static in time.

(3) The given explanation is still unsatisfactory. The initial choice of parameters in the simulations still need to be explained in the manuscript. Second, I frankly cannot follow the other explanation, even after studying the Iwata reference. Authors added: “[...] is based on the fact that in relativistic LPI the laser energy absorption occurs through collisionless mechanisms, therefore no difference is expected between the two materials. This is also the reason why collisionless PIC simulations were chosen. [...]].” Authors should explain, what the collisionless mechanisms are, why velocity throughout the layer doesn’t matter, and why it can be treated solely relativistic when the laser itself is just bordering on relativistic ($2.5e18 \text{ W/cm}^2$) and undoubtedly is comprised of a profile of intensities which dips below relativistic at the outer edges of the beam.

We disagree with the reviewer on this point. The laser energy absorption for intensities approaching, and above relativistic intensity is eminently collisionless.

The reason is that for quiver electron energies of 100’s keV and greater, the cross section for electron-ion collision at the critical density is significantly reduced, requiring a large number of laser cycles for a single collision to occur.

The hot electron generation therefore occurs through non-linear collisionless mechanisms such as resonant absorption, vacuum heating or Brunel effect, JxB acceleration, hot electron generation in magnetic channels.

(4) I think this point needs to be proven by example, particularly since the response to #3 states that 2D cannot reproduce experimental data. Also, the polarization of the laser(s) used in the experiment are not discussed in the manuscript – if they were S-polarized, this should be stated, along with the explanation written by the Authors in the rebuttal.

2D simulations cannot *quantitatively* reproduce the experimental data, however they can accurately describe the fundamental physical mechanisms at the base of our work.

We clearly specified in the manuscript that we adopted p-polarized laser light, and all simulations presented in the manuscript are performed with p-polarized laser light. We also clearly stated in the manuscript that test simulations with s-polarized laser light (not showed in the manuscript) were performed in order to demonstrate that the increment in laser energy absorption for interfering beamlets is due to vacuum-heating type mechanism and that the laser energy absorption occurs mostly along the laser polarization plane. Therefore 2D simulation provide an accurate enough picture of the relevant physics of interfering beamlets.

(5) Was convergence demonstrated?

The time-scale of collisionless absorption mechanisms near the laser critical density we are interested in this work is determined by the laser frequency/laser wavelength λ .

The time step in our PIC simulations is determined by $\Delta t < \Delta x/c$ where Δx is the cell size corresponding to $\lambda/30$. Therefore we accurately resolve the absorption physics. These are commonly used resolution values in the found in literature for laser energy absorption simulations.

(6) It is recommended to only describe the observation from the parametric study but not to make a claim that contrast doesn't play a role. Maybe a threshold could be estimated along with a corresponding time length.

Maybe there is some misunderstanding on this point. We are not claiming that the pre-formed plasma scale length does not play any role in the way laser energy is absorbed into fast electrons, but we are saying that there is enhancement of absorption over a wide range of pre-plasma scale-lengths for interfering beamlets compared to single beamlet case.

However to reply to the reviewer question, the upper boundary for this technique to be effective is when the pre-formed plasma scale-length is so large (20-30 μm) that large focal spot laser pulses will undergo multiple filamentation in the underdense plasma. In this extreme case scenario, no significant difference between single and multiple interfering beamlets will be recorded since in both cases multiple filamentation will occur. As reference we suggest the review paper by Pukhov : A. Pukhov, Reports on Progress in Physics 66, 47-101 (2003).

Our parametric study is based on short/intermediate pre-plasma scale lengths since these are those typically attained by modern, high contrast laser systems. Our mechanism is still efficient for um-scale pre-plasmas while other methods (i.e. nano-structuring), would be impractical at these contrast levels.

(7) Three explanations are offered. Wavelet self-focusing: This needs to be tied back to the coherence functions (including 2 vs 4 beam simulations). Surface modulation and Magnetic fields at the surface: Table (1) summarizes the parameter study ranging from 2.6 to 25 degrees. Experimental observations were made at the smallest angle, 2.6 degrees. If I understand Fig 3 correctly, the surface magnetic field would not play a role in the experimental configuration. Authors should tie simulations/explanations more closely to the experimental observations to be consistent with the journal's guidelines.

Maybe figure 3 is not as clear as we wished it was and we perfected it. The LFEX case of 2.6 degrees (or $11\mu\text{m}$ interference period) is the second density profile from the left. Therefore magnetic fields play a significant role in the laser energy absorption as shown in figure 4 a,b,c (corresponding to the same case), by guiding the hot electrons along the surface and injecting them in the overdense plasma at the modulation valley, where the magnetic field sign reverses.

Reviewer #2 (Remarks to the Author):

My main concern with the paper was the lack of experimental data. While the authors did not add any new data to the paper after the revision, they argued that the results they presented are reliable since the effect they observed is much stronger than what would be expected based on the level of fluctuations of laser energy and other experimental parameters. I think this is a valid claim. I also understand that it is challenging to take significant amount of shots on a kJ-level laser system, even though I am certain that an experimental run can not result in only two shots (demonstrated in the paper), so the authors should have more data. Nevertheless, I believe that the paper is important for the high-power laser community. The simulations section is very elaborate and does a great job explaining the experimental results. For these reasons I can recommend the paper for publication.

We thank the reviewer for the positive feedback and his understanding. Experimental run on kJ-class laser systems are especially challenging given the limited amount of shots available.

Reviewer #3 (Remarks to the Author):

This review is in addition to the previous two reviewers' review and their responses.

As reflected by other reviewers and the authors themselves, the manuscript presents experimental results and simulation-supported analysis of their results to derive claims that are particularly interesting to the HED community. More specifically, the authors present a significant efficiency enhancement in electron and ion conversion from the LFEX system when using angled non-coherent beamlets (compared to the same nominal energy); and identify the key players in this enhancement, namely beamlet self-focusing and converging surface magnetic fields, employing reduced dimensionality PIC simulations.

I believe that the authors have addressed relatively well the concerns raised by Reviewer 1 and 2, which were thoroughly critical in questioning the validity of the presented experimental results and conclusions. The authors have accommodated the answers to the concerns raised and have improved the overall quality of the original manuscript.

I would still like to echo the most general concerns, which are not straight-forward to address in practice, with regards to the relatively limited experimental data supporting the conclusions reached in the manuscript. While the measured ion conversion efficiency enhancement is unarguable from the experimental data alone, a more comprehensive set of experimental results would shine more light onto an otherwise fine set of simulations and resulting conclusions. Ultimately, the simulations should be used to complete the circle and validate in some of the details concerning the experimental parameters and conclusions. However, in the present form of the manuscript, the experimental results have a number of shortcomings and simulations are used to fill in relatively large voids, thus risking misinterpretation of the results. Examples of these experimental shortcomings are the lack of beamlet angle comparative results, a better characterized composite laser beam, and a more comprehensive 4-beamlet simulation.

On the other hand, one must understand the complexity of systems like LFEX and consider valid explanations for some of these shortcomings offered by the authors in their responses to reviewers. Requiring to address all raised concerns in this case, particularly in the case where they may not be practically feasible, while of utmost importance, may hamper progress and dissemination of otherwise high-quality high-impact results.

In light of these comments, I believe the manuscript is of sufficient overall quality to be considered for Nat. Comms., although it would be of unambiguous impact with supporting experimental results.

We gratefully thank the reviewer for carefully reading our paper and addressing his concerns clearly, and for giving an overall positive feedback to our work.

We completely agree with the reviewer that a larger set of data would be ideal to give full strength to the paper. However we are very confident that our experimental results are solid as mentioned in our previous reply. We are convinced that more results on the properties of interfering beamlets will soon be published from experiments on LFEX and other kJ-class facilities.

REVIEWERS' COMMENTS:

Reviewer #3 (Remarks to the Author):

In their rebuttal, the author's have clarified Reviewer 1's concerns and in some instances communicated better their findings. However, few revisions have been made to the original manuscript. I would suggest to add the following minor revisions for the purposes of clarifying terminology and methods that addresses Reviewer 1's comments:

§ A brief discussion (4-5 lines) of the low impact of dynamics wavefront distortion and how the static wavefront dominates interference and the enhancement processes related to its period etc.

§ A 2-line statement clarifying why in this regime photon absorption is collisionless (and thus is modeled as such in PIC sims). Reviewer 1 is right in that this is not clearly justified and may catch non-specialized readers off-guard

§ The authors don't actually address Reviewer 1's 5th comment. Technically, convergence is not demonstrated, as suggested in the abstract. Instead, there is evidence of convergence with simulation supported by experimental data. While this is relatively clear reading the manuscript, I would recommend a change in semantics and toning down the level of certainty expressed in the abstract with "evidence of" or "suggest that".

Reply to reviewer 3

In their rebuttal, the author's have clarified Reviewer 1's concerns and in some instances communicated better their findings. However, few revisions have been made to the original manuscript. I would suggest to add the following minor revisions for the purposes of clarifying terminology and methods that addresses Reviewer 1's comments:

§ A brief discussion (4-5 lines) of the low impact of dynamics wavefront distortion and how the static wavefront dominates interference and the enhancement processes related to its period etc.

We agree with the reviewer's suggestion, therefore we added to the text in the methods section the following:

The wave-front error in a large-scaled laser system could be classified into two kinds: static wave-front and dynamic wave-front error. The static wave-front error is related to the uniformity of the transmission optics and surface quality of both reflection and transmission optics. The dynamic wave-front error is affected by the main amplifier's thermal effect and nonlinearity (for either low energy or full energy shots), the OPCPA nonlinearity and the stability of the optics. During a full-energy shot, only thermal effects in the amplifiers and nonlinearities in the Nd:glass contributions, whose influence is limited, are added to the dynamic wave-front error. Therefore is expected that the interference pattern observed in the focal spot image shown in Figure 1b is maintained during a full energy shot.

§ A 2-line statement clarifying why in this regime photon absorption is collisionless (and thus is modeled as such in PIC sims). Reviewer 1 is right in that this is not clearly justified and may catch non-specialized readers off-guard

We agree with the reviewer and we added the following sentence to the text

The choice for collisionless simulations resides in the fact that at relativistic or near-relativistic intensities the laser energy absorption mechanisms are eminently non-collisional. In fact, for quiver electron energies exceeding few 100's keV, the cross section for electron-ion collision at the critical density is significantly reduced, thus the contribution of collisional absorption mechanisms.

§ The authors don't actually address Reviewer 1's 5th comment. Technically, convergence is not demonstrated, as suggested in the abstract. Instead, there is evidence of convergence with simulation supported by experimental data. While this is relatively clear reading the manuscript, I would recommend a change in semantics and toning down the level of certainty expressed in the abstract with "evidence of" or "suggest that".

We agree with the reviewer that technically convergence was not demonstrated for this work specifically, although detailed testing about convergence was performed when we first adopted Epoch2d as PIC code for simulations.

We also agree to tone down the level of some assertions, however the abstract was radically modified due to editorial request at this stage and we believe the present abstract to be more appropriate.